# Long Noncoding RNAs in the Pathogenesis of Insulin Resistance

**DOI:** 10.3390/ijms232416054

**Published:** 2022-12-16

**Authors:** Weili Yang, Yixiang Lyu, Rui Xiang, Jichun Yang

**Affiliations:** 1Beijing Key Laboratory of Diabetes Research and Care, Beijing Diabetes Institute, Beijing Tongren Hospital, Capital Medical University, Beijing 100730, China; 2Department of Physiology and Pathophysiology, School of Basic Medical Sciences, Peking University Health Science Center, Beijing 100191, China; 3Key Laboratory of Cardiovascular Science of the Ministry of Education, Center for Non-Coding RNA Medicine, Beijing 100191, China

**Keywords:** insulin resistance, long non-coding RNA, metabolic diseases

## Abstract

Insulin resistance (IR), designated as the blunted response of insulin target tissues to physiological level of insulin, plays crucial roles in the development and progression of diabetes, nonalcoholic fatty liver disease (NAFLD) and other diseases. So far, the distinct mechanism(s) of IR still needs further exploration. Long non-coding RNA (lncRNA) is a class of non-protein coding RNA molecules with a length greater than 200 nucleotides. LncRNAs are widely involved in many biological processes including cell differentiation, proliferation, apoptosis and metabolism. More recently, there has been increasing evidence that lncRNAs participated in the pathogenesis of IR, and the dysregulated lncRNA profile played important roles in the pathogenesis of metabolic diseases including obesity, diabetes and NAFLD. For example, the lncRNAs MEG3, H19, MALAT1, GAS5, lncSHGL and several other lncRNAs have been shown to regulate insulin signaling and glucose/lipid metabolism in various tissues. In this review, we briefly introduced the general features of lncRNA and the methods for lncRNA research, and then summarized and discussed the recent advances on the roles and mechanisms of lncRNAs in IR, particularly focused on liver, skeletal muscle and adipose tissues.

## 1. Introduction

Insulin is an anabolic hormone that is produced and released by pancreatic β cells. Following a meal or an oral glucose load, in response to elevated blood levels of glucose and other nutrients, insulin secreted by islet β cells regulates the homeostasis of glucose, lipids and protein [1]. Particularly, glucose homeostasis is dependent on insulin sensitivity of insulin target tissues, that is, a given concentration of insulin can reduce blood glucose to a normal level by targeting several organs, especially, the liver, skeletal muscle and adipose tissues [2]. Insulin suppresses endogenous glucose production (mainly hepatic glucose production), stimulates glucose utilization (mainly glycogen synthesis), inhibits lipolysis and increases lipid accumulation in the liver and adipocytes [1,3]. Insulin resistance (IR) is a pathophysiological status in which the physiological level of insulin fails to stimulate glucose uptake in adipose tissue and skeletal muscle, and suppresses hepatic glucose production, which is complicated by increased lipid metabolic disturbances in the liver and adipose tissues.

Long non-coding RNAs (lncRNAs) refers to non-protein-coding RNA molecules with transcripts longer than 200 bases. However, it has also been reported that certain lncRNAs that contain small open reading frames (smORFs, length <300 nt) can encode small peptides [4,5,6,7,8,9]. Accumulating evidence has shown that lncRNAs, which had been previously thought to have no biological functions, are widely involved in many important biological processes such as cell differentiation, body growth and development, and dysregulated lncRNA profiles are highly associated with the occurrence and development of many diseases [10,11]. So far, lncRNAs have been shown to exert their biological functions by interacting physically with DNA, RNA and proteins, either through nucleotide base pairing or via formation of structural domains induced by RNA folding [12]. The expression of lncRNA is spatiotemporally specific, and it is involved in the regulation of gene expression and biological function at epigenetic, transcriptional and post-transcriptional levels [13]. Intensive studies have revealed that there is a variety of abnormal expression patterns of lncRNAs in various tissues under insulin-resistant status, and dysregulated lncRNA profiles are associated with metabolic diseases including diabetes and nonalcoholic steatohepatitis (NASH) [14,15,16]. In this review, we briefly introduced the general features of lncRNAs and methods for lncRNA research, and then summarized and discussed the recent advances on the roles of lncRNAs in IR, particularly focused on liver, skeletal muscle and adipose tissues.

## 2. General Features of lncRNAs and Methods for lncRNA Research

### 2.1. General Features of lncRNAs

Based on the regions where they are transcribed from, lncRNAs can be classified into intergenic lncRNAs (lincRNAs), intronic lncRNAs, sense lncRNAs, antisense lncRNAs and bi-directional lncRNAs [13]. LincRNAs are transcribed from spaces between protein-coding genes, while intronic lncRNAs are transcribed from intragenic non-coding sequences. Sense lncRNAs are transcribed from the sense strand of protein-coding genes, while antisense lncRNAs are transcribed from the antisense strand. Bi-directional lncRNAs share the same promoter with protein-coding genes but are transcribed from the opposite direction [17]. According to the genomic context of lncRNAs, they also can be divided into stand-alone lncRNAs with no overlap with protein-coding genes, lncRNAs transcribed from enhancers, lncRNAs transcribed from promoters, lncRNAs transcribed from introns and antisense lncRNAs. Depending on overlapping level, antisense lncRNAs can also be further divided into divergent, terminal and nested lncRNAs [18]. Based on their relative position to adjacent genes, lncRNAs can also be divided into eight categories: divergently transcribed lncRNAs (also known as pancRNA) with two strands that share the same promoter region but transcribe along different directions, convergently transcribed lncRNAs comprised of two strands with different promoter regions but transcribe in a vis-à-vis manner, lincRNAs, overlapping antisense lncRNAs, overlapping sense lncRNAs, intronic lncRNAs and miRNA host genes [19].

Generally, lncRNAs have lower expression levels than protein-coding mRNAs and can be influenced by many factors including chromatin state, microRNAs (miRNAs) and posttranslational influences [20]. The ENCODE RNA sequencing (RNA-Seq) data of six cell lines show that most lncRNAs preferentially locate in the nucleus, especially in the chromatin region [21]. However, a number of stable lncRNAs are also located in the cytoplasm, and some are in both the nucleus and the cytoplasm [22]. Nuclear lncRNAs are mainly responsible for regulating transcription or scaffolding the nuclear compartment, while cytoplasmic lncRNAs regulate signaling transduction and post-transcriptional gene expressions [23].

LncRNAs regulate gene expressions at both transcriptional and posttranscriptional levels [24,25,26]. On transcriptional levels, lncRNAs interact with complexes that facilitate chromatin remodeling, modify the conformational change of histone by interacting with Polycomb Repressive Complex 1 and Polycomb Repressive Complex 2, resist DNA methylation through interaction with DNA methyltransferase and interact with transcription factors [19]. On posttranscriptional and epigenetic levels, lncRNAs can produce miRNAs by transcribing embedded sequences, negatively regulate miRNAs by binding miRNAs to miRNA response elements (MREs) or targeting mRNA for degradation [25,27]. LncRNAs also inhibit the functions of miRNAs by acting as miRNA sponges, which are also named as competing endogenous RNAs, by binding with miRNA binding sites such as MREs [28,29,30,31]. For example, Ghafouri-Fard et al. concluded in their review that lncRNA TMPO-AS1 served as a sponge for a number of miRNAs such as miR-320a, miR-383-5p and miR-329-3p [32]. Yu and colleagues revealed that lncRNA uc.230 increased CUG-binding protein 1 by acting as a natural sponge for miR-503 to sustain intestinal mucosal homeostasis [33]; GAS5 promotes the development of NAFLD by acting as a sponge for miR-29a-3p to enhance the expression of NOTCH2 [34]. In cytoplasm, lncRNAs also target mRNAs to alter their translation efficacy [35]. LncRNAs are also important influencers of fetal development and differentiation. The most prominent lncRNAs involved in this process include X-inactive specific transcript, H19 and maternally expressed gene 3 (MEG3). Other lncRNAs such as paired like homeodomain 2 and definitive endoderm-associated lncRNA 1 are involved in the development of the mesoderm and endoderm [19].

As mentioned above, some lncRNAs also encode peptides with biological functions. Ransohoff et al. described in their review that some lincRNAs harbour smORF-encoded micropeptides of <100 amino acids (aa) with biological function [36]. In their opinion as well as others, the lncRNAs containing these smORFs should be more properly categorized as coding rather than non-coding genes [36,37]. The authors also proposed that some loci could produce both coding and non-coding RNAs. For example, the long and protein-coding ASCC3 transcripts will switch to a short non-coding ASCC3 isoform in the presence of ultraviolet irradiation [36]. It has also been proposed in various reviews that other ncRNAs beyond lncRNAs such as circular RNAs and primary miRNAs may also encode small peptides [38,39,40] such as the small regulatory polypeptide of amino acid response (encoded by lncRNA LINC00961) [41], SHPRH-146aa (encoded by circular RNA SHPRH) [42] and miPEP-200a and miPEP171b (encoded by the primary miRNA of miR-200a and miR171b, respectively) [43,44]. Moreover, it has also been reported that HOXB-AS3 peptide but not the lncRNA HOXB-AS3 itself suppresses glucose metabolic reprogramming and tumorigenesis in colon cancer [45]. Furthermore, Bonilauri et al. also identified a subset of 15 lncRNAs containing 35 smORFs from a ribosome profiling using isolated human adipose-derived stem cells (hASC) from adipose tissue, suggesting new roles of these lncRNAs and their derived small peptides in hASC self-renewal [46]. With the advances in large-scale transcriptomes and proteomes, and the developments in bioinformatics and computational biology, more small peptides encoded by ncRNAs with important biological functions in glucose and lipid metabolism will be identified, which will definitely shed light on the roles and mechanisms of ncRNAs in glucose and lipid metabolism.

Previous studies about the roles of lncRNAs were mainly focused on their functions in hereditary disorders [47,48,49], cancers [13,50,51] and cardiovascular diseases [52,53,54,55]. Recently, large-scale transcriptome studies had revealed that many lncRNAs were involved in metabolic processes under both physiological conditions such as fasting-refeeding [56], aging [57] and adipocyte development [58], and pathological conditions such as diabetes [59,60] and fatty liver [61]. Furthermore, the relationship between lncRNAs and the development and progression of metabolic diseases including diabetes [62], nonalcoholic fatty liver disease (NAFLD) [14,63], obesity [64,65] and IR [16,66,67] had been reported. The following Section 3 is focused on summarizing and discussing the roles and mechanisms of some important lncRNAs in the pathogenesis of IR in liver, muscle and adipose tissues.

### 2.2. Methods for lncRNA Research

LncRNAs were initially considered to be the “noise” of genomic transcription. However, it has been recently revealed that lncRNAs are critical factors in the regulation of many physiological and pathological processes [68,69,70,71]. So far, many methods or techniques have been developed to study the functions of lncRNAs.

RNA-Seq is a widely applied technology that helps researchers to gain insights into the modifications of lncRNAs, such as alternative splicing and RNA editing sites via the next-generation sequencing platform [72]. RNA-seq helped to reveal the transcriptional difference between lncRNAs in the subcutaneous adipose tissue of healthy and obese women [64]. Moreover, RNA-seq is also a frequently-used method for screening disease-related lncRNAs in many diseases such as cancers [73,74], ischemic cardiomyopathy [75] and diabetes [76]. Based on the distinct characteristics of lncRNAs and mRNAs, we had developed an effective methodology named the Gene Importance Calculator to predict the importance of RNAs including lncRNAs [77], which is effective in predicting and screening important lncRNAs from the dataset when combined with microarray or sequencing analysis.

Gain- and loss-of-function experiments are also commonly used to study the function of lncRNAs. RNA interference (RNAi) is a simple and easy method to probe the quick effects of lncRNA by suppressing the expression of lncRNA using small interfering RNA (siRNA) [78,79] or much longer effects using short hairpin RNA (shRNA) [80,81] in viral vectors. Nowadays, many reports have evidence of successful RNAi knockdown of lncRNAs such as siRNA-mediated downregulation of lncRNA H19 in mouse myoblast cell line C2C12 to assess its functions in the insulin signaling of skeletal muscle [16], and shRNA-mediated knockdown of lncRNA small nucleolar RNA host gene 5 in podocytes to evaluate its functions in diabetic nephropathy [82]. Antisense oligos-mediated knockdown was also usually used in lncRNA studies. MEG3 knockdown by intravenous delivery of MEG3-locked nucleic acid-modified antisense oligonucleotides had been used to evaluate its role in NAFLD [83]. Additionally, CRISPR interference and CRISPR activation are two approaches employed to study long-term effects of lncRNAs by direct mutagenesis of lncRNAs.

The methods with the aims to characterize the interactions between lncRNAs and proteins can be categorized into protein-based approaches and RNA-based approaches. RNA immunoprecipitation (RIP) is a protein-based approach that is widely used to characterize potential RNA that has formed a complex with an indicated protein. RIP helped to validate the binding between lncRNA MALAT1 and SREBP-1c protein, and revealed its important roles in hepatic steatosis and insulin resistance [84]. Cross-linking immunoprecipitation (CLIP) is another protein-based method to study the potential protein-interacting lncRNA in living tissues and cells. Several CLIP-based databases are generated for researching lncRNA-protein interactive networks such as starbase v2.0 [85] and CLIPdb [86]. RNA-pulldown is an RNA-based approach that aims to identify the potential proteins that bind to the corresponding lncRNA. RNA-pulldown assay has been used to identify nucleolin and interleukin enhancer-binding factor 2 as a target protein that binds to the diabetic-associated lncRNA Dnm3os [87]. RNA antisense purification (RAP) as well as capture hybridization analysis of RNA targets (CHART) and chromatin isolation by RNA purification (ChIRP) are RNA-based methods designed to investigate lncRNA targets using probes according to the indicated lncRNA. RAP can also characterize a wide range of interacting elements including DNA [88,89], protein [90,91] and RNA [92,93]. ChIRP and CHART coupled with mass spectrometry or RNA-Seq are more often used to identify binding sites for lncRNAs, such as lncMALAT1 and neighbouring nuclear enriched abundant transcript (NEAT1) [94], lncDBET and fatty acid binding protein 5 [95].

To study structural–functional relationships of lncRNAs, biochemical strategies including dimethyl sulfate sequencing [96], selective 2′-hydroxyl acylation analyzed by primer extension sequencing [97], fragmentation sequencing and parallel analysis of RNA structure have also been developed [98,99,100].

## 3. LncRNAs in the Pathogenesis of IR

### 3.1. LncRNAs in Hepatic IR and Related Disease

Liver, as the key tissue of energy metabolism and switch, plays crucial roles in glucose and lipid metabolism. Liver plays a central role in regulating blood glucose homeostasis by maintaining a balance among uptake, storage and release of glucose [101]. Glucose and lipid metabolism in liver is tightly regulated by insulin, glucagon, adrenaline, growth hormones and other hormones [102]. In the feeding state, liver increases glucose uptake and converts it into glycogen and triglyceride for energy storage in response to elevated insulin level. In the fasting state, liver maintains a normal fasting blood glucose level by decomposing glycogen and increasing gluconeogenesis. In case of long-term starvation, hepatic gluconeogenesis is the main source of endogenous glucose production, which plays a decisive role in maintaining fasting blood glucose within the normal range [103,104]. However, the increase in the expression of gluconeogenic genes due to insulin deficiency or IR is the main cause of increased gluconeogenesis and fasting hyperglycemia in the pathogenesis of type 1 diabetes mellitus (T1DM) and type 2 diabetes mellitus (T2DM) [101].

Insulin-resistant liver is mainly characterized by decreased glycogen storage and triglyceride secretion with increased gluconeogenesis [105]. Mice with liver insulin receptor knockout displayed elevated expression of gluconeogenic genes, glucose intolerance and hyperglycemia [106]. So far, the mechanism of hepatic IR is still not fully understood. Recently, many lncRNAs have been shown to play important roles in the development of hepatic IR (Table 1).

#### 3.1.1. LncRNA Blnc1

Brown fat lncRNA 1 (Blnc1) is a lncRNA which regulates thermogenic adipocyte differentiation [107]. Zhao et al. reported that hepatic expression of Blnc1 was strongly elevated in high-fat diet (HFD) fed mice, and leptin-deficient (ob/ob) and leptin receptor-deficient (db/db) obese mice. Mechanistically, Blnc1 binds with endothelial differentiation-related factor 1 (EDF1), which facilitates the recruitment and coactivation of liver X receptor (LXR), leading to the activation of the LXR-sterol regulatory element-binding protein 1c (SREBP1c) pathway. Liver-specific overexpression of Blnc1 using AAV-Blnc1 under control of the liver-specific thyroid-binding globulin promoter in mice led to significantly elevated levels of triglycerides but not cholesterol in plasma and livers. Under the challenge of a NASH diet which consists of fructose, cholesterol and trans-fat, liver-specific Blnc1 knockout (KO) mice exhibited decreased liver mass and hepatic triglycerides and cholesterol content. The plasma levels of aspartate aminotransferase (AST) and alanine aminotransferase (ALT) were both downregulated in liver-specific Blnc1 KO mice. Histological tests and flow cytometry also showed reduced fibrosis and apoptotic cells in the liver tissues in liver-specific Blnc1 KO mice. [108] Overall, inactivation of Blnc1 can ameliorate the progression of NAFLD.

#### 3.1.2. LncRNA EPB41L4A-AS1

LncRNA EPB41L4A-AS1 is located in chromosome 5q22.2. Previous studies had shown that EPB41L4A-AS1 can encode a small peptide, which is named as transcript induced by growth arrest 1 (TIGA1) [109]. According to the studies by Zhang’s team, EPB41L4A-AS1 was involved in the regulation of metabolism [110,111]. In HeLa, HepG2 and L02 cells, knockdown of EPB41L4A-AS1 expression increased aerobic glycolysis and glutaminolysis [110]. Consistently, they found that in early recurrent miscarriage, upregulation of EPB41L4A-AS1 expression inhibited glycolysis in placental trophoblast cells, resulting in the inhibition of the Warburg effect [111]. They further found that EPB41L4A-AS1 showed very low expression, and was significantly negatively correlated with levels of inflammatory factors in peripheral blood mononuclear cells (PBMCs) of T2DM patients and PBMCs treated with lipopolysaccharide. Consistent with their previous findings, EPB41L4A-AS1 knockdown promoted glycolysis in PBMCs and in THP-1 cells [112]. In addition, silencing of EPB41L4AAS1 significantly increased oxygen consumption rate and enhanced mitochondrial respiration [110,111,112].

Recently, Liao et al. reported that EPB41L4A-AS1 expression was abnormally increased in the liver of patients with T2DM and up-regulated in the muscle cells of insulin-resistant patients. Consistently, EPB41L4A-AS1 expression was elevated in T2DM cell models established by a persistent high level of glucose or glucosamine treatment in human primary skeletal muscle cells, human muscle source carcinoma cells, HepG2 and L02 cells. Mechanistically, EPB41L4A-AS1 negatively regulates histone H3K27 crotonylation (H3K27cr) in the glucose transporter type 4 (GLUT4) promoter region and enhances lysine acetylation of peroxisome proliferator-activated receptor-gamma coactivator-1β (PGC-1β) through interaction with histone acetyltransferase GCN5, leading to the inhibition of GLUT4 transcription and glucose uptake. Moreover, EPB41L4A-AS1 also binds to GCN5 and enhances H3K27 and H3K14 acetylation in the thioredoxin interacting protein promoter region, enhancing GLUT4/2 endocytosis and further suppressing glucose uptake [113]. Since impaired glucose uptake is one of the major clinical features of IR and T2DM, silencing of hepatic EPB41L4A-AS1 expression seems to be a potentially effective strategy for drug development in T2DM treatment.

#### 3.1.3. LncRNA GAS5

LncRNA growth arrest-specific transcript 5 (GAS5) belongs to the GAS family, and acts as a tumor suppressor in gastric cancer, hepatocellular carcinoma, renal cell carcinoma and many other tumors by interacting with p53, phosphatase and tensin homolog (PTEN), E2F transcription factor 1 and mechanistic target of rapamycin complex 1 (mTORC1) [114,115,116,117]. The expression of GAS5 was found to negatively correlate with pancreatic and duodenal homeobox-1 (PDX1), and knockdown of GAS5 using LNA gapmeRs against GAS5 in the EndoC-βH1 cells led to impaired insulin secretion and increased apoptosis [118]

Transcriptome profiling revealed that GAS5 is upregulated in the liver tissues of HFD-induced NAFLD mice and could be reversed by metformin treatment [119]. Cui et al. also reported that the expression level of GAS5 was elevated in the liver and adipose tissues of HFD mice. GAS5-knockdown by tail vein injection of lentivirus containing shGAS5 led to GAS5 inhibition in the livers of HFD mice, accompanied with improved liver steatosis and reduced serum levels of ALT, AST, TCHO and triacylglycerol. Mechanistically, GAS5 promotes the development of NAFLD by acting as a sponge for miR-29a-3p to enhance the expression of NOTCH2 [34], which promotes lipogenesis in hepatocytes and its deregulation is related to disordered lipid metabolism in women with obesity and NAFLD [120]. In addition, Xu et al. also found that GAS5 was significantly increased in the livers of ob/ob and HFD mice, and in free fatty acids (FFAs)-treated normal human hepatocyte lines such as L02 and 7701 cells. Liver-specific GAS5 overexpression via tail vein injection of AAV8-GAS5 plasmid exacerbated hepatic lipid accumulation in HFD mice. In contrast, siRNA-mediated knockdown of GAS5 alleviated lipid accumulation and protected the mitochondrial function in FFA-treated L02 and 7701 cells. Mechanistically, GAS5 sponges miR-26a-5p to increase phosphodiesterase-4B (PDE4B) expression and subsequently promote de novo lipogenesis and damage mitochondrial function via the inhibition of cAMP/CREB pathway [121].

#### 3.1.4. LncRNA H19

H19 is a 2.4 kb paternally imprinted ncRNA, expressed from the maternal allele and is strongly expressed during embryogenesis but majorly repressed after birth, except in cardiac and skeletal muscles [122]. H19 plays important roles in regulating the functions of liver in both physiological and pathophysiological conditions [123]. Particularly, H19 was found to exert a significant role in regulating the hepatic insulin signaling cascade [124].

Goyal et al. reported that H19 was downregulated in the livers of db/db mice. In H19-silenced HepG2 cells and primary mouse hepatocytes, the mRNA and protein levels of gluconeogenic genes as well as the nuclear content of forkhead box 1 (FOXO1) were upregulated while phosphorylated insulin receptor was downregulated in the presence or absence of insulin stimulation [125]. The authors further found that H19 inhibition upregulated FOXO1 expression by increasing p53 occupancy on the FOXO1 promoter to increase its transcription [126]. Controversially, Zhang et al. reported that HFD-induced diabetic mice displayed increased hepatic expression of H19 [127], which was supported by Nilsson et al.’s report that H19 was elevated in the livers of patients with T2DM [128]. In this study, the expressions of gluconeogenic genes were increased in H19-overexpressed HepG2 cells and primary mouse hepatocytes. Liver-specific H19 overexpression by administration of AAV-H19 that expressed mouse full-length H19 elevated both the fasting blood glucose and fasting blood insulin levels, impaired glucose tolerance, promoted IR and augmented hepatic glucose production as assessed by glucose-tolerance tests, insulin-tolerant tests and pyruvate tolerance tests, respectively. In support, H19-KO mice exhibited enhanced whole-body insulin sensitivity as assessed by hyperinsulinemic–euglycemic clamp [127]. In a NAFLD mouse model induced by HFD, hepatic expression of H19 was elevated. The triglyceride levels in H19-knockdown HepG2 and Huh-7 cells were all lower than in control cells under FFA challenge. In support, silencing of H19 suppressed lipid accumulation in primary mouse hepatocytes [129]. It had been proposed that H19 promoted lipogenesis through three pathways: activating peroxisome proliferator-activated receptor γ (PPARγ) via the downregulation of miR-130a [129], activating the mTORC1 signaling axis and upregulating MLX interacting protein like (MLXIPL) [130]. Overall, although more intensive study is still needed to clarify the controversial observations regarding the roles of hepatic H19 in glucose and lipid metabolism, it is definitely involved in the pathogenesis of hepatic IR.

#### 3.1.5. LncRNA HOTAIR

LncRNA Homeobox transcript antisense RNA (HOTAIR) is located on the 12th chromosome, and is composed of seven exons. It is a crucial regulator of epigenetic modification and is upregulated in a wide variety of tumors, such as esophageal cancer, breast cancer, liver cancer, gastric cancer, renal cancer, colorectal cancer and cancer of female genitalia [131]. It is also involved in obesity-induced myocardial injury by modulating the miR-196b/IGF-1 signaling pathway [132]. Another study demonstrated that HOTAIR was highly expressed in pancreatic islets. Knockdown of HOTAIR inhibited insulin secretion by downregulating insulin transcription-related genes such as musculoaponeurotic fibrosarcoma oncogene family, protein A, PDX1 and neuronal differentiation. In addition, silencing of HOTAIR suppressed proliferation and induced apoptosis in mouse primary islets and the Min6 cell line [133].

When compared to control subjects, the expression levels of 13 ncRNAs including HOTAIR were significantly increased in PBMCs from patients with T2DM. Logistic regression analysis using T2DM as the dependent variable revealed that altered expression levels of HOTAIR and several other lncRNAs were associated significantly with T2DM, and this statistical significance still existed even after adjusting for confounding factors including age and BMI, but was lost when adjusted for HOMA-IR. This suggested that the association between HOTAIR and T2DM could be closely linked to IR [134]. Recently, Li et al. reported that the expression level of HOTAIR was significantly upregulated in the livers of T2DM patients as well as in the livers of C57BL/6J mice fed on HFD and db/db mice. In HOTAIR-overexpressed HepG2 cells, the phosphorylation levels of Akt and glycogen synthase-3-β (GSK3β) were suppressed while glucose-6-phosphatase (G6Pase) and phosphoenolpyruvate carboxykinase (PEPCK) levels were elevated. Overexpression of HOTAIR inhibited both the mRNA and protein expressions of Sirtuin-1 (SIRT1), and HOTAIR-induced upregulation of gluconeogenic genes could be reversed by SIRT1 overexpression in HepG2 cells, suggesting that HOTAIR promoted hepatic IR via the inhibition of the SIRT1 signaling pathway [135].

#### 3.1.6. LncRNA HOTTIP

LncRNA HOTTIP (HOXA transcript at the distal tip) is transcribed from the 5′-end of the HOXA gene and is located on chromosome 7p15.2, with a total length of about 4000 nt [136]. Xu et al. reported that among mouse tissues, HOTTIP was expressed most abundantly in pancreatic tissue, where it was mainly expressed in islets but not in exocrine glands. They further found that HOTTIP expression in the islets of db/db mice was downregulated when compared with that of C57BL/6J mice. HOTTIP expression was also decreased in a dose-dependent manner in Min6 cells after treatment with different concentrations of glucose for 24 h. HOTTIP knockdown with LV-shHOTTIP in Min6 cells led to the reduction of insulin secretion and cell proliferation stimulated by glucose [137]. Cao et al. reported that HOTTIP sponged miR-423-5p, which targeted wingless-type MMTV integration site family member 7A (WNT7A). HOTTIP was decreased in the liver of gestational diabetes mellitus (GDM) mice. Hydrodynamics-based plasmid overexpression of HOTTIP in GDM mouse livers ameliorated IR and hepatic gluconeogenesis via the modulation of the miR-423-5p/WNT7A axis [66].

#### 3.1.7. LncRNA LncSHGL

In one previous study, we found that in the livers of obese mice and patients with NAFLD, the expression levels of mouse lncRNA suppressor of hepatic gluconeogenesis and lipogenesis (lncSHGL), which was previously called AK14369, and its human homologous lncRNA B4GALT1-AS1 were reduced. Hepatic lncSHGL overexpression significantly attenuated global IR, hepatic gluconeogenesis and steatosis in obese diabetic mice, whereas hepatic lncSHGL inhibition promoted gluconeogenesis and lipid deposition in normal mice. An in-depth mechanistical study revealed that lncSHGL recruited heterogeneous nuclear ribonucleoprotein A1 (hnRNPA1) to enhance the translation efficiency of CALM mRNAs and increase the cellular calmodulin (CaM) protein level without affecting their transcriptions. An increase in CaM protein level activated the phosphoinositide 3-kinase (PI3K)/Akt pathway and repressed the mTOR/SREBP-1c-fatty acid synthase (FAS) pathway to suppress hepatic gluconeogenesis and lipogenesis independent of insulin and calcium in hepatocytes [138]. Clearly, inhibition of lncSHGL plays an important role in the development of hepatic IR and metabolic diseases.

#### 3.1.8. LncRNA MALAT1

Metastasis associated in lung adenocarcinoma transcript 1 (MALAT1) is a lncRNA located on the 11th chromosome primarily discovered in non-small-cell lung cancer. MALAT1 is involved in the regulation of multiple pathways such as the PI3K/Akt, MAPK/ERK and Wnt/β-cantenin pathways [139]. The serum level of MALAT1 is higher in diabetic patients than in healthy individuals [140]. MALAT1 promotes progression of diabetic complications such as neuropathy, retinopathy, nephropathy and cardiovascular diseases [141]. MALAT1 expression was increased in the diabetic neuropathy group when compared to the non-diabetic neuropathy and healthy controls groups [142]. Knockdown of MALAT1 attenuated the apoptosis of endothelial cells induced by high glucose [143].

MALAT1 expression was dose-dependently increased in HepG2 cells and primary mouse hepatocytes when exposed to different concentrations of palmitate for 24 h. Consistently, MALAT1 expression was elevated in livers of ob/ob mice. Knockdown of MALAT1 significantly decreased palmitate-induced lipid accumulation in HepG2 cells as well as in primary mouse hepatocytes, and reversed palmitate-induced nuclear SREBP-1c activation. In support, in MALAT1-overexpressed HepG2 cells, the expression of SREBP-1c was elevated. Furthermore, liver-specific MALAT1 knockdown by tail vein injection of si-MALAT1 daily for 10 days decreased blood glucose level, improved glucose intolerance, increased insulin sensitivity and decreased lipid accumulation in the liver of ob/ob mice. si-MALAT1 treatment was further shown to reduce the nuclear SREBP-1c protein level in ob/ob mouse livers. Clearly, MALAT1 induced hepatic lipid accumulation and IR by increasing SREBP-1c and its target gene expressions [84]. Another study revealed that MALAT1-KO male mice exhibited a decreased plasma insulin level and lower blood glucose level when compared to wild-type (WT) mice, which was accomplished by decreased reactive oxygen species, suppressed c-Jun N-terminal kinases 1 activity, and enhanced IRS1 and Akt phosphorylation [144]. Clearly, these studies demonstrated that MALAT1 played important roles in regulating hepatic insulin sensitivity, and inhibiting MALAT1 expression represents a potential therapeutic strategy for the treatment of IR and diabetes.

#### 3.1.9. LncRNA MAYA and ARSR

Cellular senescence plays an important role in the development of NAFLD. In HFD-fed mice, the expressions of senescence-related genes and secretory molecules were upregulated in the liver, accompanied with the increased count of senescent cells in the liver when compared to normal diet-fed ones [145]. Yes-associated protein (YAP), a major downstream effector of the Hippo signaling pathway, has been reported to be involved in regulating cellular senescence [146]. Hippo signaling interacts with Akt signaling by regulating IRS2 expression to prevent NAFLD [147]. YAP is a central regulator of glucose metabolism, controlling both enzymes involved in glycolysis and gluconeogenesis and proteins involved in glucose transport [148]. LncRNA MIST-1/2 antagonizing for YAP activation (MAYA) is a mediator of Hippo/YAP pathway. Its expression level was increased while YAP was decreased in L02 cells treated with palmitate. Inhibition of MAYA by shRNA silencing ameliorated iron overload, reduced intracellular lipid deposition as well as cellular senescence through overturning the decrease of YAP in PA-treated L02 cells. Hepatic inhibition of MAYA expression via lentivirus-shMAYA injection through the tail vein for 24 days (once per 8 days) increased hepatic YAP, accompanied by a significant amelioration of iron overload, steatosis, lipid accumulation and hepatocyte senescence in the livers of HFD mice. Overall, these data revealed that suppression of MAYA could ameliorate cellular senescence and attenuate NAFLD, likely via the upregulation of YAP and amelioration of iron overload [149].

Another lncRNA named lncRNA activated in renal cell cancer with sunitinib resistance (lncARSR) also influences the pathogenesis of NAFLD through YAP [150]. Chi et al. reported that lncARSR expression was upregulated in liver from NAFLD mice induced by HFD for 4 weeks. Consistently, lncARSR expression was elevated in HepG2 treated with 0.5 mM oleic acid. Overexpression of lncARSR by lentivirus-oe-lncARSR increased while silencing of lncARSR with lentivirus-sh-lncARSR reduced lipid accumulation and triacylglycerol contents in oleic acid-treated HepG2 cells. In addition, hepatic silencing of lncARSR in NAFLD mice via tail vein injection of lentivirus-sh-lncARSR markedly reduced lipid accumulation and triacylglycerol content in the livers. LncARSR specifically bound with YAP1 to block its phosphorylation and subsequently promote the nuclear translocation of YAP1, which activated the IRS2/AKT pathway to elevate hepatic lipid accumulation in vivo and in vitro [150]. Similarly, Zhang et al. found that lncARSR was significantly increased both in the serum and liver of NAFLD patients when compared with those of the healthy controls. LncARSR was also significantly increased in the liver of NAFLD mice induced by a methioninecholine deficient (MCD) diet when compared to chow diet fed mice. Furthermore, the expression of lncARSR was induced by FFAs, including oleic acid, palmitate and stearic acid in cultured hepatocytes. Overexpression of lncARSR promoted, while knockdown of lncARSR ameliorated lipid accumulation as well as triacylglycerol content in HepG2 cells. In accordance with the in vitro results, overexpression of lncARSR by tail vein injection with AAV8-lncARSR in the livers in MCD mice significantly increased hepatic lipid accumulation, while inhibition of hepatic lncARSR by tail vein injection with AAV8-shlncARSR obviously decreased hepatic lipid accumulation. The authors further proposed that lncARSR promoted hepatic steatosis by activating the PI3K/Akt/mTOR/SREBP-1C pathway [151].

#### 3.1.10. LncRNA MEG3

LncRNA MEG3 is a maternally imprinted gene located on the 14th chromosome and contains ten exons. *MEG3* is an oncogenic lncRNA downregulated in many malignancies including breast cancer, liver cancer, colorectal cancer, cervical cancer, gastric cancer and ovarian cancer [152]. Recent studies showed that MEG3 was also involved in the development of hepatic IR.

The expression of MEG3 was significantly elevated in the liver tissues of mice fed on HFD and ob/ob mice. In addition, palmitate treatment time dependently increased MEG3 expression in mouse primary hepatocytes [153,154]. Hepatic MEG3 knockdown by injecting si-MEG3 fragments via the tail vein twice a week for 10 weeks improved glucose intolerance and IR in HFD mice. Mechanistically, the authors found that upregulation of hepatic lncRNA MEG3 promoted hepatic IR by increasing FOXO1expression [153]. They further showed that MEG3 functioned as a competing endogenous RNA of miR-214, and MEG3 knockdown led to the upregulation of miR-214 and inhibited its target gene activating transcription factor 4 expression. This repressied the expression of FOXO1 and its downstream gluconeogenic enzymes G6pase and PEPCK to suppress gluconeogenesis and ameliorate IR [110]. Similarly, another study confirmed that MEG3 level was up-regulated in palmitate-treated HepG2 cells and primary hepatocytes, and in HFD mouse livers. MEG3 knockdown via sh-MEG3 transfecting reversed the downregulation of Akt and GSK3β induced by palmitate while MEG3 overexpression via cDNA-MEG3 transfecting showed the opposite effect in HepG2 cells and primary hepatocytes. In HFD C57BL/6J mice, hepatic MEG3 knockdown via tail vein injection of sh-MEG3 markedly downregulated the body weight, blood glucose and serum insulin levels, accompanied by improved glucose tolerance [155]. Cheng et al. reported that MEG3 expression was elevated in human livers with NAFLD and NASH. However, contrary to the above studies, MEG3 knockdown by intravenous delivery of locked nucleic acid-modified antisense oligonucleotides against MEG3, which led to 94.8%, 89.8% and 68.9% reduction of MEG3 expression in the liver, skeletal muscle and epididymal white adipose tissue (eWAT), respectively, did not improve insulin sensitivity but exacerbated overall glucose intolerance and IR in mice fed on HFD for 10 weeks. MEG3 knockdown reduced the phosphorylated Akt level in liver and skeletal muscle tissues but not in eWAT in obese mice. MEG3 knockdown also reduced the level of phosphorylated insulin receptor β subunits (IRβ) in the liver. Overall, these findings suggested that MEG3 knockdown impaired insulin sensitivity in the liver and skeletal muscle [83].

Controversial to the findings that MEG3 was upregulated in IR model described above, in a recent study conducted by Zou et al., the expression of MEG3 was time dependently downregulated in primary hepatocytes after treatment with FFAs. MEG3 was also reduced in liver tissue of HFD-fed mice and NAFLD patients. MEG3-overexpressed primary hepatocytes exhibited decreased lipid accumulation and triglyceride content. Hepatic MEG3 overexpression via tail vein injection of MEG3-expressing AAV for 12 weeks in HFD mice also ameliorated obesity and hepatic lipid deposition. They further revealed that MEG3 directly interacted with enhancer of zeste homolog 2 (EZH2) to destabilize it through ubiquitin-mediated degradation and subsequently upregulate sirtuin 6 (SIRT6), a target for EZH2, suppressing lipogenesis and hepatic inflammation [156]. SIRT6 suppresses carbohydrate response element binding protein (ChREBP), SREBP1 and LXRα through deacetylation, and further inhibits their downstream proteins to repress hepatic lipogenesis [157]. Activating the MEG3-EZH2-SIRT6 axis significantly suppressed lipid accumulation and inflammation in vitro, and ameliorated NAFLD in vivo [156]. Clearly, more in-depth studies are needed to elucidate the distinct roles and mechanisms of MEG3 in the pathogenesis of hepatic IR and metabolic diseases.

#### 3.1.11. LncRNA NONMMUT031874.2

By a high-throughput sequencing analysis, Zhang et al. found that hepatic lncRNA NONMMUT031874.2 was upregulated in HFD mice. Metformin treatment ameliorated hepatic IR by downregulating hepatic NONMMUT031874.2 expression and increasing the expression of miR-7054-5p to inhibit its target gene, suppressor of cytokine signaling 3 (SOCS3). Inhibition of SOCS3 finally increased insulin sensitivity to activate PI3K/AKT pathway, suppressing hepatic gluconeogenesis [158].

**Table 1 ijms-23-16054-t001:** LncRNAs in hepatic IR and related disease.

LncRNAs	Pathology	Model	Hepatic Expression	Potential Treatment Strategies	Reference
Blnc1	obesity and hepatic steatosis	HFD, ob/ob and db/db mice	elevated	Inactivation of Blnc1 to restrain EDF1/LXR/SREBP-1C pathway	[108]
EPB41L4A-AS1	T2DM	T2DM patients and T2DM cell models	elevated	Silencing of EPB41L4A-AS1 to restore the inhibited GLUT4 transcription and elevated GLUT4/2 endocytosis	[113]
GAS5	obesity and NAFLD	ob/ob and HFD-induced NAFLD mice, and FFA-treated hepatocyte lines	elevated	Inhibiting of GAS5 to upregulate miR-29a-3p and suppress the expression of Notch2 or upregulate miR-26a-5p to suppress the expression of PDE4B and activate cAMP/CREB pathway	[34,121]
H19	T2DM	db/db mice	decreased	Upregulating of H19 to inhibit FOXO1 expression	[126]
H19	T2DM	HFD-induced diabetic mice and patients with T2DM	elevated	Inhibiting of H19 to suppress lipogenesis via inactivating PPARγ and mTORC1 signaling axis and downregulating MLXIPL	[127,128,129,130]
HOTAIR	T2DM	T2DM patients and HFD and db/db mice	elevated	Inhibition of H19 to promote SIRT1 signaling pathway	[134,135]
HOTTIP	gestational diabetes mellitus (GDM)	GDM mice	decreased	Overexpression of HOTTIP ameliorated IR and hepatic gluconeogenesis via the modulation of the miR-423-5p/WNT7A axis	[66,137]
LncSHGL	obesity and NAFLD	obese mice and patients with NAFLD	decreased	Activation of lncSHGL recruited hnRNPA1 to upregulate CaM, which activates PI3K/Akt pathway and represses mTOR/SREBP-1C pathway	[138]
MALAT1	obesity	ob/ob mice and palmitate-induced HepG2 cells	elevated	Inhibiting MALAT1 to decrease SREBP-1c and its target gene expressions	[84,144]
MAYA	NAFLD	FFA-treated hepatocyte lines and HFD mice	elevated	Suppression of MAYA attenuated NAFLD via the upregulation of YAP and subsequent amelioration of iron overload	[149]
ARSR	NAFLD	oleic acid-induced hepatocyte lines, HFD mice and NAFLD patients	elevated	Suppression of ARSR attenuated hepatic steatosis via the phosphorylation and subsequently block the nuclear translocation of YAP1, and inactivation of the IRS2/AKT pathway	[150,151]
MEG3	obesity and NAFLD	HFD and ob/ob mice palmitate-induced hepatocyte	elevated	Suppression of MEG3 improved hepatic IR by decreasing FOXO1 expression	[153,154,155]
MEG3	NAFLD	NAFLD patients	elevated	MEG3 knockdown impaired insulin sensitivity in the liver via the inhibition of Akt and IRβ	[83]
MEG3	NAFLD	HFD mice, NAFLD patients and NAFLD patients	decreased	Activating the MEG3-EZH2-SIRT6 axis suppressed lipid accumulation and inflammation in vitro, and ameliorated NAFLD in vivo	[156]
NONMMUT031874.2	NAFLD	HFD mice	elevated	Inhibition of NONMMUT031874.2 repressed SOCS3 to increase insulin sensitivity and activate PI3K/AKT pathway	[158]

### 3.2. LncRNA in Muscular IR and Related Disease

Skeletal muscle plays a crucial role in glucose disposal, and it accounts for about 30% of glucose utilization in physiological conditions and about 80% of insulin-stimulated whole-body glucose uptake in healthy individuals [159,160]. In individuals with T2DM, insulin-stimulated glucose disposal deceases by at least half mainly due to reduction in glucose uptake of skeletal muscle [161]. Clearly, skeletal muscle is one of the primary targets of IR, and muscular IR contributes much to the development of T2DM and other metabolic diseases [162].

Insulin-resistant muscles have lower glucose uptake when stimulated with insulin, and display impaired fatty acid oxidation. Muscular IR is mainly characterized by impairment of GLUT4 translocation and glucose uptake, reduction of glycogen synthesis and tyrosine phosphorylation of IRS1 under insulin stimulation [163]. Recently, RNA profiling analysis revealed that in insulin-resistant skeletal muscle cells induced by palmitate treatment, the expressions of multiple lncRNAs were altered, suggesting that lncRNAs were involved in the pathogenesis of muscular IR [164,165,166,167] (Table 2).

#### 3.2.1. LncRNA H19

Recent studies have linked the dysregulation of H19 to cardiometabolic diseases and alterations in cell metabolism [168]. Particularly, H19 was also implicated in muscular IR. Long-term paternal exercise could reduce the expression of H19 and alter several metabolic genes including GLUT4, FOXO1 and pyruvate dehydrogenase kinase 4, in offspring skeletal muscle in C57BL/6J male mice, and offspring mice whose fathers were exposed to a long-term exercise regimen exhibited lower energy expenditure and an increased risk of obesity and insulin resistance on a HFD [164,169]. However, it had been also reported that paternal exercise improved the metabolic health of offspring via epigenetic modulation of the DNA methylation profile of PI3Kca, H19, and insulin-like growth factor 2 (IGF2) in the skeletal muscle of the offspring in mice [165].

Gao et al. reported that H19 level was downregulated in the skeletal muscle of HFD mice and diabetic human subjects. Given that H19 can function as a sponge for miRNA let-7, downregulation of H19 increased the bioavailability of let-7 without affecting its transcription. This reduces the expressions of let-7 target genes, including INSR and lipoprotein lipase, to exacerbate the pathogenesis of IR and T2DM. Moreover, acute hyperinsulinemia downregulated H19 through PI3K/AKT-dependent phosphorylation of the miRNA processing factor KH domain-containing AU-rich element binding protein, which also promotes biogenesis of let-7 to mediate H19 destabilization. Clearly, these findings revealed a double-negative feedback loop between sponge lncRNA H19 and the target miRNA let-7, and upregulation of H19 blocked this vicious cycle and restored glucose metabolism disorder in muscular cells [170]. Furthermore, Amit Kumar et al. reported that the H19 level was also significantly downregulated in the skeletal muscles of db/db mice. Decrease in H19 expression activated histone deacetylase 6 (HDAC6) in skeletal muscle cells, leading to the decreased expression of IRS1 and impaired insulin signaling [16]. Injection of adenovirus-H19 via tail vein resulted in overexpression of H19 in skeletal muscle and liver with improved global IR [171]. Inhibition of H19 by siRNA transfection impaired glucose uptake in mouse myotubes under insulin stimulation [166]. Moreover, H19 KO mice exhibited decreased global insulin sensitivity. It had been proposed that H19 also regulated muscular insulin signaling by interacting with hnRNPA1, which increased translation efficacy of fatty acid oxidation genes including peroxisome proliferator-activated receptor-gamma coactivator-1α (PGC-1α) and carnitine palmitoyltransferase 1B (CPT1B) [171], or modulating the activity of adenosine 5′-monophosphate (AMP)-activated protein kinase (AMPK) [166]. Overall, these studies suggested that activation of H19 could be potential strategy to improve muscle IR and T2DM.

#### 3.2.2. LncRNA IRLnc

Intramuscular fat (IMF) refers to the fat distributed in skeletal muscle fibers. Excessive accumulation of IMF has been reported to be associated with muscular IR and diabetes in humans [172,173]. An RNA sequencing using longissimus dorsi muscle tissue from high- and low-IMF pigs revealed that six lincRNAs exhibited significantly different expression. Among them, one lincRNA named IMF related lincRNA (IRLnc), which was highly conserved across 100 vertebrates and humans, was significantly increased in muscle tissue of high-IMF pigs compared to low-IMF pigs. Among genes upstream and downstream of IRLnc, nuclear receptor subfamily 4 group A member 3 (NR4A3) was the only gene that displayed differential expression in the muscle tissue between high-IMF pigs and low-IMF pigs. IRLnc silencing significantly decreased the RNA and protein expressions of NR4A3 in cultured primary myoblasts. In situ hybridization analysis and lincRNA–RNA interaction prediction suggested that IRLnc may affect NR4A3 expression by directly binding to its 3′-UTR region. The authors further proposed that IRLnc directly promoted the expression of NR4A3 to inhibit catecholamine catabolism and finally promote IMF deposition, causing muscular IR [174].

#### 3.2.3. LncRNA NONMMUT044897.2

Administration of resveratrol, a drug mainly harvested from the plant *Polygunum cuspidatum* with anti-oxidative and anti-inflammatory effects, ameliorated global and muscular IR with the alterations of 338 mRNAs and 629 lncRNAs in skeletal muscles of HFD mice [167]. Among the altered lncRNAs, NONMMUT044897.2 was the most upregulated lncRNA in the skeletal muscles of the HFD mice, but this was reversed after resveratrol treatment. Resveratrol ameliorated IR in mouse skeletal muscles by downregulating NONMMUT044897.2 expression, increasing the expression of miR-7051-5p to reduce the expression of its target gene, suppressor of cytokine signaling 1 (SOCS1). Finally, inhibition of SOCS1 increased insulin sensitivity to activate the PI3K/Akt pathway, enhancing glucose uptake in skeletal muscles [167].

**Table 2 ijms-23-16054-t002:** LncRNAs in muscular IR and related disease.

LncRNAs	Pathology	Model	Muscular Expression	Potential Treatment Strategies	Reference
H19	T2DM and NAFLD	HFD and db/db mice and diabetic human subjectsv	decreased	Activation of H19 could be potential strategy to improve muscle IR and T2DM via multi pathways such as blocking H19/let-7 vicious cycle, inactivating histone HDAC6 to upregulate IRS1 and interacting with hnRNPA1 to increase fatty acid oxidation genes	[16,166,170,171]
IRLnc	muscular IR and diabetes	muscular IR pigs	elevated	IRLnc directly promoted the expression of NR4A3 to inhibit catecholamine catabolism and finally promote IMF deposition, causing muscular IR	[174]
NONMMUT044897.2	muscular IR	HFD mice	elevated	Downregulating NONMMUT044897.2 expression to increase miR-7051-5p and inhibit SOCS1, increasing insulin sensitivity in skeletal muscles	[167]

### 3.3. LncRNA in Adipocyte IR and Related Disease

Adipose tissue, which includes WAT, brown adipose tissue (BAT) and beige adipose tissue, is an endocrine and vital energy storage organ that regulates whole-body energy metabolism [175,176,177]. Adipocytes not only play a role in lipid storage, but also on metabolism and inflammation via the secretion of adipokines such as leptin, resistin, adiponectin, tumor necrosis factor-alpha (TNFα) and interleukin (IL)-6 [178,179,180]. Insulin stimulates glucose uptake in adipocytes. Upon binding with insulin receptor on cell membranes, insulin activates PI3K and downstream molecule Akt, which leads to the phosphorylation of Akt substrate 160 kDa (AS160), which subsequently activates Rab10. Rab10 then interacts with RAS-like oncoprotein A and membrane-to-cortex linker Myosin-Ic to stimulate the translocation of GLUT4 from cytoplasm to membrane, enhancing glucose uptake [181]. After uptake by GLUT4, glucose enhances lipogenesis by activating ChREBP. Insulin also inhibits lipolysis by suppressing phosphodiesterase 3b, phosphorylating adipose-specific phospholipase A2 and suppressing FOXO1 and interferon regulatory factor 4. Insulin also stimulates the lipid uptake of adipose tissue from blood, reducing the plasma fatty acid level [182].

Insulin-resistant adipose tissue displays impaired glucose and triglyceride uptake under insulin stimulation, leading to hyperglycemia and hyperlipidemia. In insulin-resistant WAT, the activity of Akt and translocation of GLUT4 from cytoplasm to membrane are significantly blunted [183]. Recently, there has also been increasing attention to the roles of lncRNAs in the occurrence and development of IR in adipocyte, in which lncRNAs function as powerful regulators of adipocyte differentiation and gene expressions [184]. When compared with the Sham group, Roux-en-Y gastric bypass significantly improved IR with marked change in the expressions of 87 lncRNAs in the WAT of obese diabetic rats. Among the 87 altered lncRNAs, 49 lncRNAs were upregulated while 38 lncRNAs were downregulated [185]. Salvianolic acid B (Sal B), as a proven anti-obesity substance, can significantly reduce the body mass, the weight of body fat and serum levels of several lipids such as triglycerides, low-density lipoprotein cholesterol and total cholesterol. In the eWAT of HFD-induced obese mice, the expressions of 234 lncRNAs were altered after treatment with Sal B. Among 234 altered lncRNAs, 87 lncRNAs were upregulated and 147 lncRNAs were downregulated. The upregulated lncRNAs were mainly involved in lipid transport and metabolism, while the downregulated lncRNAs were involved in immunologic or inflammatory responses [186]. Clearly, these findings revealed that lncRNAs played important roles in regulating insulin signaling and metabolism in adipose tissues (Table 3).

#### 3.3.1. LncRNA ADINR and NEAT1

A group of lncRNAs including SRA, HOTAIR, adipogenic differentiation induced noncoding RNA (ADINR) and NEAT1 that are involved in the generation of white adipose tissue, are involved in the pathogenesis of obesity [187]. SRA and HOTAIR can promote preadipocyte differentiation, and global knockout of SRA protects against HFD-induced obesity in mice [188]. ADINR interacts with CCAAT/Enhancer binding protein α subunit (C/EBPα) to enhance adipogenesis. C/EBPα is a required factor for white adipose tissue differentiation, and it exerts adipogenic actions by inducing and maintaining PPARγ levels [189]. NEAT1 is significantly upregulated in mature adipocytes when compared to adipocyte-derived stem cells. Adipocyte-derived stem cells isolated from miR-140 knockout mice had dramatically decreased adipogenic capabilities associated with the downregulation of NEAT1 expression, and overexpression of NEAT1 in miR-140 knockout adipocyte-derived stem cells was sufficient to restore their ability to undergo differentiation with unknown mechanisms [187]. More functional assays are required to further probe the role of NEAT1 in promoting adipogenesis [187].

#### 3.3.2. LncRNA ADIPINT

Kerr A et al. reported that human adipocyte-specific lncRNA (CATG00000106343.1), which was also renamed as adipocyte specific pyruvate carboxylase interacting RNA (ADIPINT), was increased in WAT from obese women when compared that of non-obese women. Moreover, subcutaneous adipose (sWAT) ADIPINT expression correlated positively with body mass index, body fat, adipocyte cell volume and plasma triglyceride levels. Its expression also correlated with overall in vivo HOMO-IR and in vivo adipose IR. ADIPINT inhibition decreased the abundance and enzymatic activity of pyruvate carboxylase in the mitochondria, inhibited adipocyte lipid synthesis and reduced lipid content. The authors proposed that ADIPINT was a regulator of lipid metabolism in human WAT by binding to pyruvate carboxylase and increasing its abundance and enzymatic activity, subsequently stimulating triglyceride synthesis, increasing lipid droplet size and promoting adipose IR in adipocytes [190].

#### 3.3.3. LncRNA ASMER-1 and ASMER-2

Comparison between the lncRNA profile of subcutaneous and visceral WAT from insulin sensitive and insulin resistant obese women revealed 44 differentially expressed lncRNAs. LncRNA ENSG00000235609.4 and CATG00000111229.1 were chosen for further study, and were named as adipocyte specific metabolic related lncRNA 1 and 2 (ASMER-1 and ASMER-2), respectively, because they exhibited adipose-enriched expressions. Moreover, their expressions were altered in the conditions of obesity and IR. ASMER-2 was upregulated while ASMER-1 was downregulated in the adipose tissue of obese and insulin resistant women. Interestingly, very similar results were obtained following the silencing of two lncRNAs in differentiated human adipocytes in vitro. Silencing of ASMER-1 and ASMER-2 similarly inhibited lipolysis and adiponectin release in adipocytes, although ASMER-2 antisense oligos resulted in a stronger repressive effect than ASMER-1 antisense oligos [191].

#### 3.3.4. LncRNA Blnc1

Blnc1, previously called as AK038898, was identified by Zhao et al. as a driver of thermogenesis in brown and beige adipocytes in 2014 and so it was renamed [192]; it participates in the regulation of metabolic pathways involving obesity [193]. The authors declared that Blnc1 formed a ribonucleoprotein complex with transcription factor early B cell factor 2 (EBF2) to stimulate the expression of a thermogenic gene such as UCP1. Moreover, EBF2 also enhanced the expression of the Blnc1 itself, thereby creating a reinforcing loop to promote adipogenesis toward thermogenic phenotype [194]. They further found that Blnc1 was highly conserved in mice and humans at the sequence level as well as functionally conserved in its regulation of brown adipocyte gene expression. Blnc1also physically interacted with heterogeneous nuclear ribonucleoprotein U (hnRNPU), which interacted with EBF2 to facilitate the formation of EBF2/Blnc1 ribonucleoprotein complex [195]. Furthermore, the authors revealed that zinc finger and BTB domain containing 7b (Zbtb7b), a transcription factor, was required for brown fat development and thermogenesis. Mechanistically, Zbtb7b physically binds with hnRNPU, and then recruits the hnRNPU/Blnc1 ribonucleoprotein complex to stimulate thermogenic gene expression in adipocytes [196]. The expression of Blnc1 was elevated in eWAT and BAT of HFD-induced obese mice and ob/ob mice. The expression of Blnc1 in eWAT was strongly associated with weight gain and eWAT mass during HFD feeding. Adipocyte-specific Blnc1-knockout (AKO) mice displayed an increased glucose level, insulin level, glucose intolerance and IR as well as more serious hepatic steatosis after HFD feeding. In contrast, fat-specific Blnc1 transgenic mice (Blnc1 Tg mice) exhibited much lower plasma insulin concentrations, improved insulin sensitivity and glucose tolerance as well as ameliorated hepatic steatosis. The mRNA expression of UCP1 in BAT and the mRNA expressions of genes involved in de novo lipogenesis (SREBP-1C, FAS and stearoyl-CoA desaturase), lipid storage (diacylglycerol acyltransferase 2 (DGAT2), fatty-acid-binding protein 4 and cell death-inducing DFF45-like effector C) and catabolism (enoyl-CoA, hydratase/3-hydroxyacyl CoA dehydrogenase and Acetyl-Coenzyme A acyltransferase 1B) were significantly reduced while the mRNA levels of several macrophage markers, including Galectin 3, C-type lectin receptor 4D, carboxypeptidase A3 and TNFalpha-induced protein 2 were elevated in the eWAT of HFD-fed AKO mice. HFD-fed Blnc1 Tg mice exhibited the opposite effects. Together, these results demonstrated that adipose-specific Blnc1 inactivation or transgenic expression exacerbated or alleviated diet-induced systemic IR and hepatic steatosis by promoting or suppressing BAT whitening and eWAT inflammation, respectively. A proposed mechanism was that Blnc1 acted cooperatively with its protein partner Zbtb7b to attenuate proinflammatory cytokine signaling and promote fuel storage in adipocytes. Elevated Blnc1 in adipose tissue of HFD and ob/ob mice was a protective mechanism against obesity-induced brown fat whitening, adipose tissue inflammation and fibrosis [197]. In support of the benefit roles of Blnc1 in adipose, a study from another team showed that overexpression of Blnc1 in eWAT by multi-point injection of adenovirus carrying Blnc1 into eWAT on both sides of HFD-induced obese mice ameliorated glucose intolerance, hepatic steatosis and systemic IR and attenuated mitochondrial dysfunction as evidenced by increased mitochondrial mass and expression of genes related to mitochondrial functions. In contrast, knockdown of Blnc1 led to mitochondrial dysfunction in 3T3-L1 pre-adipocytes. Blnc1 stimulated the transcription of PGC-1β via binding with hnRNPA1 [198]. However, contrary to the roles of Blnc1 in adipose, Xie et al. reported Blnc1 played a pro-inflammatory role in primary brain microvascular endothelial cells and mediated the permeability and inflammatory response of cerebral hemorrhage by activating the PPARγ/SIRT6/FoxO3 pathway [107]. Consistent with the study of Xie et al., Feng et al. reported that inhibition of Blnc1 could reduce the fibrosis, inflammation and oxidative stress in high glucose-induced HK-2 cells, and Blnc1 promoted inflammation, oxidative stress and renal fibrosis by activating the Nrf2/HO-1 and NF-κB pathways in diabetic nephropathy [199].

Coupled with the effects of promoting NASH in the liver as described above, the current studies revealed that Blnc1 played different roles in different human diseases; therefore, its function is complicated and may be far from clinical application.

#### 3.3.5. LncRNA Dio3os

Maternal obesity predisposes female offspring metabolic dysfunction and obesity [200]. Chen et al. found that maternal obesity negatively affected BAT functions primarily in female offspring but barely affected the male offspring. Dio3 antisense RNA (Dio3os) is a maternally imprinted lncRNA that was suppressed in BAT of maternal obesity female fetuses and neonates [201,202]. To specifically overexpress Dio3os in BAT, AAV-Dio3os were locally injected into interscapular BAT of female neonate mice. After a 5-week feeding on a normal diet at 22 °C, AAV-Dio3os-treated mice displayed less body weight gain and higher BAT mass, consistent with increased body energy expenditure, higher oxygen consumption and core body temperature after exposure to cold climate, and improved glucose sensitivity. Mechanistically, Dio3os suppression in offspring BAT activated Dio3, which reduced thyroid hormone triiodothyronine action, leading to impaired activity of PRD1-BF1-RIZ1 homologous domain containing 16 (PRDM16) and impaired thermogenesis [203].

#### 3.3.6. LncRNA GAS5

It had been reported that adipose stem cells from obese patients showed specific differences in Gas5 [204]. In contrast to the increase in liver of HFD-induced NAFLD mice, serum GAS5 level was reduced in diabetic patients [205,206]. Luo et al. also reported that the serum level of GAS5 was significantly lower in patients with T2DM when compared with health control subjects. They also found that a low serum level of GAS5 was associated with high levels of HbAlc, fasting glucose and LDL-c in patients with T2DM. The serum level of GAS5 could be restored in patients treated with oral hypoglycemic agents or insulin [207]. They also found that both insulin content and insulin secretion were increased in GAS5-overexpressed INS-1 832/13 cells [207]. In adipocytes from T2DM patients, the expression of GAS5 was also significantly decreased. GAS5 was found to bind to the promoter of insulin receptor and stimulate its transcription, enhancing insulin signaling to increase glucose uptake in adipocytes [206].

#### 3.3.7. LncRNA Gm15290

LncRNA Gm15290 has been reported to promote cell proliferation and invasion in non-small cell lung cancer [208,209]. Recently, Liu et al. identified 2246 differentially expressed lncRNAs with fold change greater than 2.0 by comparing the lncRNA expression profiles in the WAT of WT mice and ob/ob mice using lncRNA microarray analysis. Among the deregulated lncRNAs, lncRNA Gm15290 was one of the most significantly upregulated lncRNAs in WAT of ob/ob mice. Gm15290 overexpression increased, whereas Gm15290 knockdown reduced lipid accumulation in mouse primary preadipocytes after induced differentiation for 8 days. Consistently, the expression of key adiogenic genes including the master gene PPARγ, early adiogenic marker gene C/EBPa and late adiogenic marker gene aP2 were markedly induced by Gm15290 overexpression but were suppressed by Gm15290 knockdown. The authors further found that Gm15290 regulated miR-27b, a miRNA that played important roles in both adipocyte differentiation and adipocyte insulin resistance through targeting insulin receptors [210] and PPARγ [211], to promote PPARγ-induced fat deposition in WAT, resulting in body weight gain in mice [212].

#### 3.3.8. LncRNA H19 and MEG3

DNA methylation of IGF2/H19 has been shown to be associated with adiposity and metabolic diseases [213]. It had been reported that H19 expression was lower while MEG3 expression was higher in the sWAT of obese women when compared to those of normal-weight women. H19 expression had an inverse correlation while MEG3 expression had a positive correlation with obesity indices and HOMA-IR values in humans. Furthermore, H19 expression displayed an inverse correlation with FAS while a positive correlation was found between MEG3 and FAS and PPARγ mRNA levels in SAT [214]. After the administration of agrin H19-gain-of-function (AGR-H19-Rgof) mimics via intraperitoneal injection, which led to a substantial proportion of AGR-H19-Rgof mimics in skeletal muscle and a minimal accumulation of that in the liver, kidneys, lungs and hearts, HFD mice displayed lower body weight, increased lean weight and reduced fat weight without significant influence on food intake or physical activity. The serum cholesterol and triglyceride levels were also reduced, there was also decreased lipid accumulation in the liver and WAT in treated mice. Ob/ob mice treated with AGR-H19-Rgof also showed decreased body mass, increased muscle mass and decreased lipid deposition. In addition, Elena Schmidt et al. reported H19 increased upon cold-activation and decreased in obesity in BAT, and had an inverse correlation with BMI in humans. H19 overexpression promoted, and H19 silencing impaired adipogenesis, oxidative metabolism and mitochondrial respiration in brown but not white adipocytes [215]. Overall, these data suggested a potential anti-obesity effect of H19 [216].

#### 3.3.9. LncRNA Linc-ADAL

LincRNA for adipogenesis and lipogenesis (linc-ADAL) is abundantly expressed in human adipose. Linc-ADAL expression could be activated by PPARγ in human adipocytes, and its expression is upregulated in adipose depots of obese patients when compared to that of lean humans. It was also markedly induced in vitro during hASC differentiation to adipocytes. Antisense oligonucleotides-mediated knockdown of linc-ADAL in hASC preadipocytes markedly impaired subsequent adipocyte differentiation, reduced triglyceride accumulation and inhibited the expressions of lipid biosynthetic genes such as SREBF1 and FAS in differentiated adipocytes. Mechanistically, linc-ADAL interacted with hnRNPU and insulin-like growth factor 2 mRNA binding protein 2 (IGF2BP2) at distinct subcellular locations to promote adipocyte differentiation and lipogenesis [217].

#### 3.3.10. LncRNA LncASIR

Adipose-specific insulin responsive lncRNA (lncASIR) is an insulin-responsive lncRNA mainly located in the nucleus. Under insulin stimulation, the expression of lncASIR in adipocytes is notably increased when compared to control cells. LncASIR decreased by 50% after overnight fasting whereas it showed a more than 10-fold increase after HFD feeding in inguinal white adipose tissue (iWAT), eWAT and BAT. This was consistent with a reduced insulin level during fasting but an increased insulin level in the HFD condition. After silencing lncASIR in cultured primary adipocytes using the dcas9 system, several genes including DGAT2, ATP citrate lyase, thyroid hormone-responsive, AcCoA synthase short chain family member 2, 1-acyl-sn-glycerol-3-phosphate acyltransferase 2 and aldehyde dehydrogenase 3B2, which were related to peroxisome proliferator-activated receptor signaling, lipolysis and adipocytokine signaling were downregulated. The altered metabolic profile of lncASIR-knockdown adipocytes illustrated a role of lncASIR in regulating insulin signaling transduction in adipose [218].

#### 3.3.11. LncRNA LncOb

Recently, Dallner O et al. reported that LncOb, another fat-specific lncRNA, was expressed exclusively in white adipocytes, and its expression was reduced in WAT of fasted mice but increased in that of HFD mice. LncOb expression was markedly increased when preadipocytes isolated from eWAT differentiated into adipocytes, accompanied with a significant elevation in leptin mRNA level. In addition, knockdown of lncOb using two different locked nucleic acids (LNAs) all led to significant reduction in leptin expression in primary adipocytes. LncOb KO mice exhibited increased body weight and fat mass with a significantly decreased plasma leptin level. Consistent with these observations in lncOb KO mice, large-scale genetic studies of humans revealed a significant association of single-nucleotide polymorphisms in the region of human lncOb with a lower plasma leptin level, higher BMI and body fat percentage as well as increased risk of extreme obesity. The authors further found that lncOb bound to the leptin proximal promoter sequence and promoted its transcription [219]. It was also reported that the lncOb rs10487505 variant was associated with increased levels of plasma leptin in youths with NAFLD, and was also associated with body weight and plasma leptin reduction after bariatric surgery [220]. These data provided evidence that variation in human lncOb RNA could lead to hypoleptinemia and obesity.

#### 3.3.12. LncRNA SRA

LncRNA steroid receptor RNA activator (SRA) is a coactivator of steroid receptors as well as an interactor with multiple kinds of protein involved in malignancies such as colorectal cancer, ovarian cancer, endometrial cancer and osteosarcoma [221,222,223,224,225]. In adipose tissue of non-diabetic obese patients, the expression of SRA1 was significantly elevated when compared to that of normal weight individuals. However, SRA1 expression in adipose tissue differed non-significantly between diabetic and non-diabetic participants. In non-diabetic participants, adipose tissue SRA1 expression was associated directly with BMI, percentage of body fat, waist circumference, fasting serum insulin and HOMA-IR while only associated inversely with the HbA1c in individuals with T2DM. These data indicated that SRA1 may have potential as a biomarker of metabolic disorders [226]. Xu et al. reported that SRA expression was induced during 3T3-L1 preadipocytes differentiated into mature adipocytes, and SRA was highly expressed in white adipose tissue. SRA bound to PPARγ and enhanced its transcriptional activity. SRA overexpression promoted the differentiation of adipocyte precursor ST2 cells while SRA knockdown inhibited the differentiation of 3T3-L1 preadipocytes. Affymetrix GeneChip microarray analysis in SRA overexpressing ST2 adipocytes and SRA knockdown 3T3-L1 adipocytes revealed that SRA regulated insulin sensitivity and insulin-stimulated glucose uptake, and inhibited TNFα signaling in adipocytes [227]. A follow-up study from this team found that SRA expression was significantly increased in WAT from HFD-induced obese mice when compared with control mice fed normal chow. They generated a global mouse Sra1 gene knock-out (SRA global KO, SRAKO) mice to assess SRA function in vivo. SRAKO mice exhibited similar body weight changes as WT mice when fed a normal chow diet, but were resistant to HFD-induced obesity with a reduced eWAT and sWAT mass, and reduced liver mass. Consistent with the reduced fat mass after HFD feeding, SRAKO mice had significantly reduced insulin levels with no change in fasting blood glucose when compared with the WT mice. Furthermore, HFD-fed SRAKO mice exhibited less severe glucose intolerance as well as IR than WT mice [188]. Overall, these data suggested that anti-SRA may be a potential strategy for obesity-induced IR and obesity.

#### 3.3.13. LncRNA TUG1

The expression of lncRNA taurine upregulated gene 1 (TUG1) was downregulated in both SAT and visceral adipose tissue of obese women [228]. Zhang et al. reported that overexpression of TUG1 via tail intravenous injection with TUG1 overexpressing lentivirus ameliorated glucose and insulin intolerance, attenuated fatty accumulation and suppressed inflammation in testicular adipose tissues in HFD-induced obese mice. Mechanistically, TUG1 inhibited the expression of miR-204, leading to the upregulation of SIRT1, one target gene of miR-204. Upregulation of SIRT1 activated the expressions of GLUT4, PPARγ and the AKT pathway to suppress inflammation and improve insulin sensitivity in adipocytes [229]. In another study from this research group, they revealed that TUG1 was significantly downregulated in the adipose tissues from diabetic mice, and the injection of TUG1 overexpression lentivirus via the tail vein significantly attenuated obesity and serum glucose levels, and ameliorated the accumulation of testicular adipose tissue in diabetic mice. Furthermore, the overexpression of TUG1 significantly increased SIRT1, ATGL, PPARα, PGC-1α and UCP-1 expression and decreased the expression of miR-204 in adipose tissues and high glucose-induced 3T3-L1 preadipocytes [230]. Based on these findings, they proposed that lncRNA TUG1 promoted the browning of white adipose tissue by regulating miR-204/SIRT1 axis to ameliorate diabetes [229,230].

#### 3.3.14. LncRNA uc001kfc.1

LncRNA uc001kfc.1 is highly expressed in human adipose tissue [231]. Whole transcriptome analysis revealed uc001kfc.1 and PTEN expressions were lower in the adipose tissue from an obese group when compared with that of z control group [232]. Interestingly, the authors also reported that the mRNA level of FOXO1 as well as fasting blood glucose were significantly lower in the obese group, and they even found enhanced insulin sensitivity in obese adipose tissue which was in conflict with previous findings that obesity led to impaired glycometabolism and IR [233,234,235,236]. They attributed this puzzling phenomenon to the differences in the pathological stage and the metabolic status of selected subjects. The core network analysis of obesity predicted uc001kfc.1 as a potential regulatory lncRNA of PTEN. Thereby, they proposed that decreased uc001kfc.1 might be linked to the down-regulation of PTEN, which increased PIP3 to phosphorylate and activate AKT to enhance insulin sensitivity of white adipocytes in obese patients [232].

**Table 3 ijms-23-16054-t003:** LncRNAs in adipocyte IR and related diseases.

LncRNAs	Pathology	Model	WAT Expression	Potential Treatment Strategies	Reference
ADINR	NA	NA	NA	ADINR interacted with CEBPα to enhance adipogenesis	[187]
NEAT1	obesity	miR-140 knockout mice	decreased	Overexpression of NEAT1 in miR-140 knockout adipocyte-derived stem cells was sufficient to restore their ability to undergo differentiation with unknown mechanisms	[202]
ADIPINT	obesity	WAT from obese women and non-obese women	elevated	ADIPINT bound to pyruvate carboxylase and increase its abundance and enzymatic activity, subsequently stimulating triglyceride synthesis, increasing lipid droplet size and promoting adipose IR in adipocytes	[190]
ASMER-1	Obesity and IR	obese and insulin resistant women	decreased	Silencing of ASMER-1 inhibited lipolysis and adiponectin release in adipocytes	[191]
ASMER-2	Obesity and IR	obese and insulin resistant women	elevated	Silencing of ASMER-2 inhibited lipolysis and adiponectin release in adipocytes	[176]
Blnc1	Obesity and IR	HFD mice and ob/ob mice	elevated	Adipose-specific Blnc1 transgenic expression alleviated diet-induced systemic IR and hepatic steatosis by suppressing BAT whitening and eWAT inflammation through several targets including EBF2, hnRNPU, Zbtb7b, hnRNPA1 and PGC-1β	[194,195,196,197,198]
Dio3os	maternal obesity and IR	HFD-induced maternal obesity and maternal obesity female/male offspring	NA but suppressed in BAT of maternal obesity female fetuses and neonates	BAT-specific activating Dio3os activated brown adipogenesis and thermogenesis, improving energy expenditure and glucose intolerance	[203]
GAS5	T2DM	T2DM patients	decreased	GAS5 bound to the promoter of insulin receptor and stimulated its transcription, enhancing insulin signaling to increase glucose uptake in adipocytes	[206]
Gm15290	Obesity	ob/ob mice and mouse primary preadipocytes	elevated	Gm15290 regulated miR-27b but not miR-27c to promote PPARγ-induced fat deposition in WAT, resulting in body weight gain	[212]
H19	Obesity	obese and normal-weight women, HFD mice and ob/ob mice	decreased	Overexpression of H19 using AGR-H19-Rgof mimics reduced body weight, increased lean weight and decreased fat weight in HFD and ob/ob mice	[214,215,216]
MEG3	Obesity	obese and normal-weight women	elevated	MEG3 expression displayed a positive correlation with FASN and PPARγ mRNA levels in SAT	[214]
Linc-ADAL	Obesity	obese patients	elevated	Linc-ADAL interacted with hnRNPU and IGF2BP2 at distinct subcellular locations to promote adipocyte differentiation and lipogenesis	[217]
LncASIR	Obesity	HFD mice	elevated	Silencing of lncASIR led to the suppression of PPAR signaling, lipolysis and adipocytokine signaling in cultured primary adipocytes	[218]
LncOb	Obesity	HFD mice and youths with NAFLD	elevated	LncOb bound to the leptin proximal promoter sequence and promoted its transcription, and activating lncOb in adipose tissue may be effective in improving obesity	[219,220]
SRA	obesity	non-diabetic obese patients and HFD-induced obese mice	elevated	SRA overexpression promoted the differentiation of adipocyte precursor ST2 cells while SRA knockdown inhibited the differentiation of 3T3-L1 preadipocytes via bound to PPARγ and enhanced its transcriptional activity.	[188,226,227]
TUG1	Obesity and T2DM	obese women, obese mice and diabetic mice	decreased	LncRNA TUG1 promoted the browning of white adipose tissue by regulating miR-204/SIRT1 axis to ameliorate diabetes	[229,230]
uc001kfc.1	obesity	obese women	decreased	Decreased uc001kfc.1 might be linked to the down-regulation of PTEN, which increased PIP3 to phosphorylate and activate AKT to enhance insulin sensitivity of white adipocytes in obese patients	[232]

NA: means not mentioned in the reference.

## 4. Conclusions and Perspectives

IR is caused by a combination of genetic and environmental factors, and is one of the earliest manifestations of a constellation of human diseases including T2DM, NAFLD and cardiovascular disease. It has been found that a set of lncRNAs play important roles in regulating the insulin signaling pathway in liver, adipose and muscle tissues (Figure 1 and Figure 2 and Table 1, Table 2 and Table 3). A dysregulated lncRNA profile is widely involved in the pathogenesis of IR and metabolic diseases. In the past decade, studies on the roles and mechanisms of lncRNAs in regulating glucose and lipid metabolism have greatly expanded our understanding on the insulin signaling pathway, IR and metabolic diseases, and provided a number of potential targets for the diagnosis and treatment of diabetes, NAFLD and other metabolic diseases (Figure 2). Targeting lncRNA represents a novel strategy for the treatment of metabolic diseases.

In the future, more intensive study is still needed to probe the in-depth roles and mechanisms of reported lncRNAs in regulating glucose and lipid metabolism in various tissues beyond liver, adipose and muscle. Particularly, given the huge number of lncRNAs in humans, it is also crucial to identify and study new lncRNAs with important roles in regulating glucose and lipid metabolism.

## Figures and Tables

**Figure 1 ijms-23-16054-f001:**
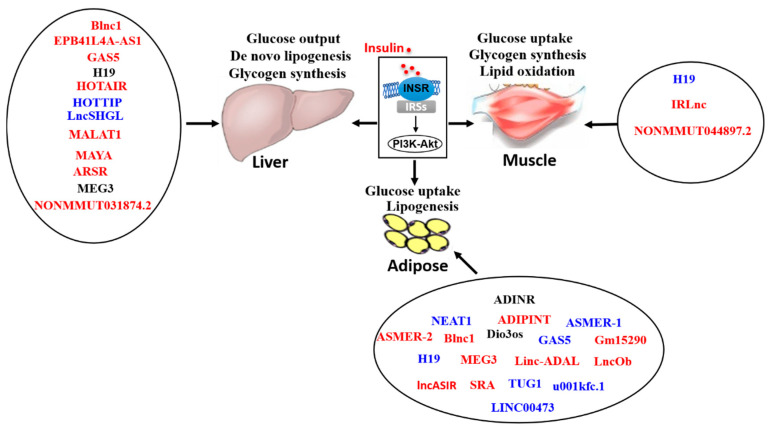
A deregulated lncRNA profile is involved in the pathogenesis of insulin resistance and dysregulated glycolipid metabolism in various tissues. Under the condition of metabolic disorders, the declined expression of lncRNAs are marked in blue, and the increased expression of lncRNAs are marked in red, while those with unknown or controversial expression are marked in black in liver, skeletal and adipose tissues. Generally, upregulated lncRNAs are negative regulators of the insulin signaling pathway, while downregulated lncRNAs exert beneficial effects on insulin sensitivity and glycolipid metabolism. INSR, insulin receptor; IRSs, insulin receptor substrates.

**Figure 2 ijms-23-16054-f002:**
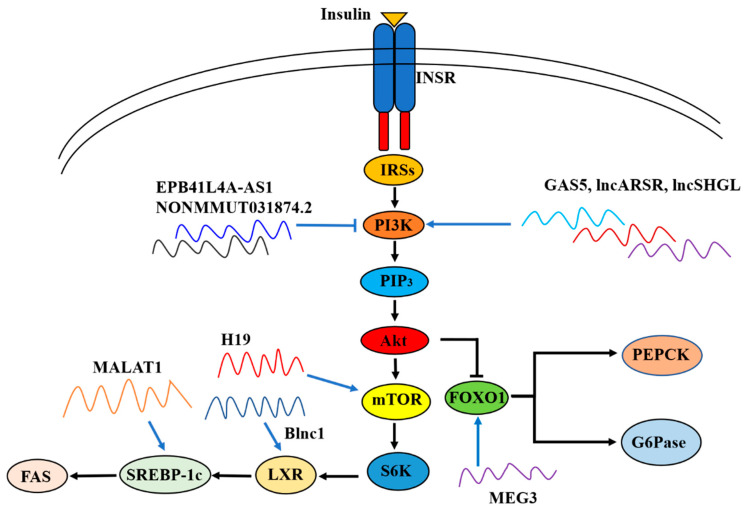
LncRNAs are important regulators of the insulin signaling pathway and glycolipid metabolism in the liver. LncRNAs including GAS5, lncARSR and lncSHGL positively regulate the PI3K/Akt pathway, while lncRNAs including EPIB41L4A-AS1 and NONMMUT031874.2 negatively regulate the PI3K/Akt pathway in the liver. Other lncRNAs interact with downstream molecules of the PI3K/Akt pathway to regulate glucose and lipid metabolism: H19 activates mTOR complex, MEG3 stimulates FOXO1 and its downstream gluconeogenic enzymes, Blnc1 activates the LXR/SREBP-1c pathway, and MALAT1 upregulates the expression of SREBP-1c to promote hepatic gluconeogenesis and lipid deposition. Akt, protein kinase B; FAS, fatty acid synthase; FOXO1, forkhead box o1; G6Pase, glucose-6-phosphatase; INSR, insulin receptor; IRSs, insulin receptor substrates; LXR, liver x receptor; mTOR, mammalian target of rapamycin; PEPCK, phosphoenolpyruvate carboxykinase; PI3K, phosphatidylinositol 3-kinase; PIP3, phosphatidylinositol (3,4,5)-triphosphate; S6K, ribosomal protein S6 kinase; SREBP-1c, sterol regulatory element binding protein-1c.

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
