# Peer review of "Long Noncoding RNAs in the Pathogenesis of Insulin Resistance"

_ijms, 2022, doi:10.3390/ijms232416054_

Round 1

Reviewer 1 Report

Overall, this is a comprehensive review of the status of lncRNAs in NAFLD and IR.

It should be noted that not all NAFLD is equivalent to IR. Perhaps the title should be altered to include both since the review embodies both quite extensively. Note that this is alluded to in the conclusion on p 21 (just before Fig. 1).

The methodology section 2.1, focuses disproportionately on cancer and other pathways wherein lncRNAs have been implicated. This section could be shortened considerably as this is not a methodology review. In addition, it would be helpful if the review mentioned methodology papers that have been used in insulin resistance (although we recognize that much of the technology has evolved in genetics and cancer research).

In fact, section 2.2 would be better placed earlier in the review (ahead of section 2.1). It will be easier to understand the methods if the features of lncRNAs are described first.

In section 3.1.1. The sentence “Mechanistically, Blnc1…(SREBP1c) pathway.” Would be better earlier in the paragraph after the first sentence. It is helpful to know the function and mechanism of a lncRNA prior to discussion of its changes effected by the knockout or overexpression.

In addition to the overall “graphic abstract” showing all the lncRNAs generically impacting the liver and metabolism, it would be helpful if the authors provided some figures depicting the more specific pathways of the lncRNAs they deem best characterized vis a vis IR. This would enhance the text, especially for individual readers who are interested in IR but are not necessarily molecular biologists.

Section 3.3.5. Please relate the findings here re: Dio3os with IR. Obesity, while a risk for T2D, is not the subject of this review.

Edits of grammar and English usage would benefit the readability of the review.

Some abbreviations are used only once or are nonstandard (e.g., NeuroD, RM, DN, HC, MO etc.) and should be eliminated as there are plenty of acronyms already in the manuscript. As a general rule, only use an abbreviation/acronym if used > 3 times. Palmitate, oleic acid, stearic acid, etc. should not be abbreviated; excessive abbreviations make the manuscript unnecessarily distracting to read and keep track of abbreviations.

Author Response

Response to Reviewer 1:

Overall, this is a comprehensive review of the status of lncRNAs in NAFLD and IR. It should be noted that not all NAFLD is equivalent to IR. Perhaps the title should be altered to include both since the review embodies both quite extensively. Note that this is alluded to in the conclusion on p 21 (just before Fig. 1).

Responses: We thank a lot for the constructive comment. As requested, we had altered the title in the section 3.1 to “LncRNAs in hepatic IR and related disease”; the title in the section 3.2 to “LncRNA in muscular IR and related disease”; the title in the section 3.3 to “LncRNA in adipocyte IR and related disease”.

The methodology section 2.1, focuses disproportionately on cancer and other pathways wherein lncRNAs have been implicated. This section could be shortened considerably as this is not a methodology review. In addition, it would be helpful if the review mentioned methodology papers that have been used in insulin resistance (although we recognize that much of the technology has evolved in genetics and cancer research).

Responses: We greatly thank you for these very constructive comments. As requested, we had shortened this section to a certain extent. In addition, we had also replaced some studies in genetics and cancer with studies in insulin resistance and metabolic diseases accordingly (New section 2.2).

In fact, section 2.2 would be better placed earlier in the review (ahead of section 2.1). It will be easier to understand the methods if the features of lncRNAs are described first.

Responses: Thanks a lot for this important suggestion. As requested, section 2.2 (General features of lncRNAs) has been placed ahead to be the new section 2.1, and hence the section (Methods for lncRNA research) come to be the new section 2.2.

In section 3.1.1. The sentence “Mechanistically, Blnc1…(SREBP1c) pathway.” Would be better earlier in the paragraph after the first sentence. It is helpful to know the function and mechanism of a lncRNA prior to discussion of its changes effected by the knockout or overexpression.

Responses: We thank you for this important suggestion. As requested, the sentence “Mechanistically, Blnc1…(SREBP1c) pathway.” has been placed ahead in the paragraph after the first sentence in section 3.1(LncRNA Blnc1).

In addition to the overall “graphic abstract” showing all the lncRNAs generically impacting the liver and metabolism, it would be helpful if the authors provided some figures depicting the more specific pathways of the lncRNAs they deem best characterized vis a vis IR. This would enhance the text, especially for individual readers who are interested in IR but are not necessarily molecular biologists.

Responses: Thanks a lot for this important suggestion. As suggested, a new figure (figure 2) had been provided to summarize the key mechanisms of some important lncRNAs in the regulation of insulin signaling pathways and glucose/lipid metabolism in the liver beyond previous figure 1.

Section 3.3.5. Please relate the findings here re: Dio3os with IR. Obesity, while a risk for T2D, is not the subject of this review.

Responses: Thanks a lot for this constructive comment. It is true that Dio3os is just related to obesity according to the data in the study cited here in section 3.3.5. So, to make our description more precisely, we had altered the title in the section 3.1 to “LncRNAs in hepatic IR and related disease”; the title in the section 3.2 to “LncRNA in muscular IR and related disease”; the title in the section 3.3 to “LncRNA in adipocyte IR and related disease”.

Edits of grammar and English usage would benefit the readability of the review.

Responses: Thanks for the reviewer's comment. The manuscript has been thoroughly revised for correcting grammar issues by us and other senior professors.

Some abbreviations are used only once or are nonstandard (e.g., NeuroD, RM, DN, HC, MO etc.) and should be eliminated as there are plenty of acronyms already in the manuscript. As a general rule, only use an abbreviation/acronym if used > 3 times. Palmitate, oleic acid, stearic acid, etc. should not be abbreviated; excessive abbreviations make the manuscript unnecessarily distracting to read and keep track of abbreviations.

Responses: We thank you for pointing out this issue. As suggested, the abbreviations had been thoroughly revised throughout the text and the tables, and marked in red.

Reviewer 2 Report

This review article provided an overview on the roles of long non-coding RNAs (lncRNAs) in insulin resistance (IR), particularly focused on liver, skeletal muscle and adipose tissues.

This review certainly needs to be improved, in fact, the description of the role of lncRNAs as endogenous competitors in miRNA sponges is missing, even if this aspect was mentioned in the introduction, it needs to be deepened for each lncRNA described in the review.

The authors state that some lncRNAs encode for small polypeptides but there is no experimental evidence for this, therefore evidence should be provided, or it should be reported that even if in the presence of possible open reading frames these are not converted into proteins.

The abstract is too general and does not provide an overview of the review.

Author Response

Response to Reviewer 2:

This review article provided an overview on the roles of long non-coding RNAs (lncRNAs) in insulin resistance (IR), particularly focused on liver, skeletal muscle and adipose tissues. This review certainly needs to be improved, in fact, the description of the role of lncRNAs as endogenous competitors in miRNA sponges is missing, even if this aspect was mentioned in the introduction, it needs to be deepened for each lncRNA described in the review.

Responses: We greatly thank the expert for this important suggestion. As requested, we had added the description of the roles of lncRNAs as endogenous sponge for miRNA in the new section 2.1. Moreover, we had also provided more in-depth mechanistical description of lncRNAs in the manuscript.

The authors state that some lncRNAs encode for small polypeptides but there is no experimental evidence for this, therefore evidence should be provided, or it should be reported that even if in the presence of possible open reading frames these are not converted into proteins.

Responses: We greatly thank the expert for this important comment. As requested, we had added a new paragraph to describe the discovery of some lncRNAs that encode small polypeptides in the new section 2.1.

The abstract is too general and does not provide an overview of the review.

Responses: Thanks a lot for the constructive comment. As suggested, we had revised the abstract with a more comprehensive overview of the review.

Round 2

Reviewer 2 Report

The authors have improved the manuscript which can be published in this new version.